# EXTREME TENSORING FOR LOW-MEMORY PRECON-DITIONING

**Xinyi Chen**
Google AI Princeton
Princeton, NJ
`xinyic@google.com`

**Naman Agarwal**
Google AI Princeton
Princeton, NJ
`namanagarwal@google.com`

**Elad Hazan**
Princeton University & Google AI Princeton
Princeton, NJ
`ehazan@cs.princeton.edu`

**Cyril Zhang**
Princeton University & Google AI Princeton
Princeton, NJ
`cyril.zhang@princeton.edu`

**Yi Zhang**
Princeton University
Princeton, NJ
`y.zhang@cs.princeton.edu`

## ABSTRACT

State-of-the-art models are now trained with billions of parameters, reaching hardware limits in terms of memory consumption. This has created a recent demand for memory-efficient optimizers. To this end, we investigate the limits and performance tradeoffs of memory-efficient adaptively preconditioned gradient methods. We propose *extreme tensoring* for high-dimensional stochastic optimization, showing that an optimizer needs very little memory to benefit from adaptive preconditioning. Our technique applies to arbitrary models (not necessarily with tensor-shaped parameters), and is accompanied by regret and convergence guarantees, which shed light on the tradeoffs between preconditioner quality and expressivity. On a large-scale NLP model, we reduce the optimizer memory overhead by three orders of magnitude, without degrading performance.

## 1 INTRODUCTION

Among the most influential and important optimization techniques in machine learning are adaptive learning-rate methods, otherwise known as diagonal-matrix adaptive preconditioning. Essentially all of the most-commonly used incarnations of adaptive preconditioning (AdaGrad, Adam, RMSprop, Adadelta, etc.) accumulate second-moment estimators of each coordinate of the gradient, then scale the parameter updates by the square roots of these accumulators. These methods come with an overhead memory cost of storing these accumulators, thereby doubling the memory consumption. In the regime where model size encroaches upon the same order of magnitude as the total amount of RAM, a need has arisen to view memory as a limited resource in large-scale optimization.

We address the question of whether the benefits of adaptive preconditioning can be attained without significant memory overhead. To this end, we introduce *extreme tensoring*, a family of generic modifications to any second-moment-based adaptive optimizer. Our method uses a compressed preconditioner which takes the form of a tensor product of arbitrary order, with simple updates, without necessarily requiring the parameters to be tensor-shaped. In our regret analysis, we quantify how extreme tensoring competes provably with full-memory AdaGrad in the online convex optimization framework, with a multiplicative data-dependent constant that can be measured empirically.

In a large-scale language modeling setting, we demonstrate that an optimizer requires very little additional memory to benefit from adaptive preconditioning. Furthermore, the inherent flexibility

of our method enables us to conduct, to the first of our knowledge, the first empirical study of the tradeoff between training convergence and memory *in the optimizer*.

## 1.1 RELATED WORK

The most widely-adopted form of adaptive preconditioning are second-moment-based: for example, AdaGrad (Duchi et al., 2011), Adam (Kingma and Ba, 2014), RMSprop (Tieleman and Hinton, 2012), and Adadelta (Zeiler, 2012). Some recent preconditioning methods are not based on second moments (Bello et al., 2017; Chen and Gu, 2018; Bernstein et al., 2018), and fall beyond our scope.

**Tensor-factorized preconditioners.** Many works (Martens and Grosse, 2015; Gupta et al., 2018; Martens et al., 2018) investigate tensor-factorized preconditioners. These are presented in the view of restricted full-matrix preconditioning (vs. diagonal for us) of tensor parameters (vs. general-purpose for us). In parameter count regimes relevant to this paper, these full-matrix adaptive methods suffer from prohibitive time and memory overhead issues. Extreme tensoring can be viewed as the diagonal version of Shampoo. The matrix version of diagonal Shampoo was discussed in the appendix of (Gupta et al., 2018); in this paper we extend the algorithm to arbitrary tensor factorizations, and focus on the tradeoff between memory consumption and optimizer performance. The theoretical part of our work follows proof techniques seen in (Gupta et al., 2017) and (Gupta et al., 2018), although our diagonal restriction results in distinct updates and incomparable regret bounds.

**Optimizer memory reduction in deep learning.** Perhaps the most closely related work to ours is Adafactor (Shazeer and Stern, 2018), an empirical work which achieves sublinear-memory adaptive regularization by restricting preconditioners on matrix-shaped gradients to "row" and "column" learning rates, with a similar update rule. Similar algorithms have appeared in prior work (Gupta et al., 2014; Shazeer et al., 2017). As a special case of our regret analysis, we provide some theory for this line of work, while proposing a more general and versatile memory-reduction method. Concurrent to this work, Anil et al. (2019) are also motivated by memory efficiency, and the algorithm uses a different approximation to the preconditioner than a tensor product. Their algorithms' index partitions are compatible with our tensor indices, and we provide a comparison in Appendix C.

**Optimizer memory reduction outside of deep learning.** Older than our work is the classic L-BFGS algorithm (Liu and Nocedal, 1989), a quasi-Newton method with a tunable memory window hyperparameter. This does not immediately apply to our setting, as the tunable memory overhead is (window size)$\times$, rather than $(1 + \varepsilon)\times$. Furthermore, second-order methods turn out to be brittle under stochastic gradients; it remains an open empirical problem to robustly exploit Hessian curvature of the loss surface in modern deep learning settings.

**Intrinsic dimensionality in deep learning.** More broadly, there are various empirical attempts to locate and characterize low-dimensional phenomena hiding within high-dimensional parameter spaces in deep learning (Li et al., 2018a;b). Our comparison provides a lens on measuring the intrinsic dimensionality of the *preconditioner*. Another is (Agarwal et al., 2020), a different way to impose parameter bottlenecks on adaptive preconditioners; however, the proposed method does not reduce memory consumption.

**Our contributions:**

- Using the user-selectable degree of memory reduction, we conduct an empirical study of training convergence vs. preconditioner memory usage. We find, in a realistic large-scale language modeling setting, that reallocating memory savings from the optimizer to the model itself can pay off in terms of end-to-end performance.
- We propose extreme tensoring as a modification to AdaGrad, Adam, etc. for reducing the overhead memory cost in adaptive preconditioning, accompanying algorithms such as (Shazeer and Stern, 2018; Anil et al., 2019).
- We derive a regret bound for extreme-tensored AdaGrad, and provide the first numerical comparisons of regret bounds in deep learning. Though they obviously do not provide convergence guarantees in the non-convex setting, they act as a proxy for understanding the AdaGrad accumulator overestimation induced by memory constraints. Via this diagnostic lens, we see that the regret competes with AdaGrad within a multiplicative constant, which we measure to be small in practice.

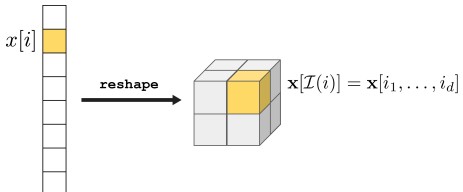

Figure 1: Illustration of *tensor index* $\mathcal{I}$: the vector $x$ is reshaped into tensor $\mathbf{x}$ of $d$ dimensions, and the bijection $\mathcal{I}$ maps the $i$th coordinate of $x$, $x[i]$, to $\mathbf{x}[\mathcal{I}(i)]$.

## 2 PRELIMINARIES

### 2.1 STOCHASTIC OPTIMIZATION AND ADAPTIVE METHODS

We will state our algorithmic contribution in the usual framework of stochastic optimization of a differentiable function $f(\cdot)$, equipped with an unbiased stochastic gradient oracle $\widetilde{\nabla} f(\cdot)$. At each of $T$ iterations, the algorithm queries a stochastic gradient $g_t := \widetilde{\nabla} f(x_t)$. In practice, our target is the large-scale ERM setting, where $f$ is the population loss on training data, and $\widetilde{\nabla} f$ is a mini-batch stochastic gradient with respect to the model parameters.

However, since the introduction of AdaGrad (Duchi et al., 2011), the standard analysis of adaptive regularization has been in the online convex optimization (OCO) framework (see, e.g. Gupta et al. (2017; 2018)). In this language, we view the $x_t$ queried by the learning algorithm as a sequence of online *decisions* chosen from a convex set $\mathcal{K}$, and the $g_t$ as a sequence of gradients of adversarially chosen convex loss functions $f_t$. We are interested in minimizing the regret, defined as

$$\sum_{t=1}^{T} f_t(x_t) - \min_{x \in \mathcal{K}} \sum_{t=1}^{T} f_t(x).$$

This is a generalization of stochastic convex optimization; for a broad survey, see Hazan (2016). A framework for reducing the stochastic *non-convex* setting to the online convex setting is introduced in Agarwal et al. (2018), translating a sublinear regret bound to convergence to approximate stationary points.

### 2.2 TENSORS AND INDEXING CONVENTIONS

We introduce some notation and terminology for clarity and convenience.

By $[n]$ we denote the index set $\{1, \ldots, n\}$. $\mathbf{I}_d$ and $\mathbf{0}_d$ refer to the $d \times d$ identity matrix and $d$-dimensional zero vector, respectively. We use $\bullet$ to denote the trace inner product (Hadamard product) between two matrices, and $\cdot$ to denote the element-wise product of two vectors, or a vector and a scalar.

As is standard in optimization contexts, a tensor of order $p$ is an element of $\mathbb{R}^{d_1 \times \ldots \times d_p}$: a $p$-dimensional table of real numbers, indexed by a $p$-tuple $I = (I_1, \cdots I_p)$ of integers, where $I_i \in [d_i]$. Our methods will require the user to specify a relabeling of gradient vectors, which we will call a *tensor index*:

**Definition 2.1.** *Let $d_1, \ldots, d_p$ be positive integers whose product is $d$. A* tensor index *is a bijection $\mathcal{I} : [d] \to \bigtimes_{i=1}^{p} [d_i]$ between indices for $\mathbb{R}^d$ and $\mathbb{R}^{d_1 \times \ldots \times d_p}$.*

We will refer to the "reshape" conversion between vectors $x \in \mathbb{R}^d$ and tensors $\mathbf{x} \in \mathbb{R}^{d_1 \times \ldots \times d_p}$ specified by the index bijection $\mathcal{I}$. For this, we will use the shorthand notation $\mathbf{x} := \mathcal{I}(x)$ and $x := \mathcal{I}^{-1}(\mathbf{x})$. Throughout this paper, we will use square brackets to refer to vector and tensor indices: for example, the previous definition can be restated as enforcing $x[i] = \mathbf{x}[\mathcal{I}(i)]$. See Figure 1 for an illustration of such a conversion.

Although we do not need them to state the simple algorithm, it will be useful in the analysis to introduce a few other pieces of notation. We denote the tensor (or Kronecker) product of matrices

$A, B$ by $A \otimes B$. Given a positive definite matrix $A \in \mathbb{R}^{d \times d}$, for any $x \in \mathbb{R}^d$, define the matrix norm $\|x\|_A := \sqrt{x^\top A x}$ and its dual $\|x\|_A^* := \sqrt{x^\top A^{-1} x}$. For a square matrix $M$, $\mathrm{diag}(M)$ refers to the diagonal matrix of the same dimensions and diagonal entries as $M$.

## 3   PRECONDITIONING BY EXTREME TENSORING

We begin by presenting our proposal for black-box memory reduction in adaptive preconditioning. We call this *extreme tensoring*, as it takes the view of restricting a high-dimensional preconditioner to an arbitrary-order tensor product.

The algorithm implicitly maintains a preconditioner which is a rank-one tensor product of the same dimension as a given tensor index. A second-moment accumulator is maintained for sums of squared gradient magnitudes across each $(p-1)$-dimensional tensor slice; the adaptive step size for each coordinate is the inverse square root of the geometric mean of its $p$ corresponding slice sums. The formal specification of the base algorithm (AdaGrad with extreme tensoring) is given in Algorithm 1.

---

**Algorithm 1** AdaGrad with extreme tensoring

1: **Input:** Initializer $x_1$, learning rate schedule $\{\eta_t\}$, tensor index $\mathcal{I}$ with dimensions $(d_1, \ldots, d_p)$, $\varepsilon > 0$
2: Initialize $(S^{(1)}, \ldots, S^{(p)}) := (\mathbf{0}_{d_1}, \ldots, \mathbf{0}_{d_p})$
3: **for** $t = 1, \ldots, T$ **do**
4:     Receive stochastic gradient $g_t$
5:     Reshape: $\mathbf{g}_t := \mathcal{I}(g_t)$
6:     Accumulate slice sums:
$$\forall i \in [p], \quad S^{(i)}[j] \leftarrow S^{(i)}[j] + \sum_{I:I_i=j} \mathbf{g}_t[I]^2$$
7:     Get step sizes: $\delta_t[I] := (\varepsilon + \prod_{i=1}^p S^{(i)}[I_i])^{-\frac{1}{2p}}$
8:     Update: $x_{t+1} \leftarrow x_t - \eta_t \cdot \mathcal{I}^{-1}(\delta_t) \cdot g_t$
9: **end for**

---

We make a few important remarks:

- AdaGrad is a special case of Algorithm 1, with $p = 1, d_1 = d$. The analogues for Adam, RMSprop, Adadelta, etc. are obtained straightforwardly by decaying the accumulator ($S^{(i)}[j] \leftarrow \beta_2 \cdot S^{(i)}[j] + \ldots$). Extreme tensoring is compatible with first-moment estimation (i.e. momentum), although the memory savings disappear. In the setting for our main empirical result, removing momentum did not degrade performance, corroborating the findings of Shazeer and Stern (2018).

- If the tensor dimensions $(d_1, \ldots, d_p)$ in the decomposition are close to equal, the memory overhead scales as $O(p \cdot d^{1/p})$.

- As is standard in matrix and tensor optimization methods (Gupta et al., 2018; Shazeer and Stern, 2018; Martens et al., 2018), independent copies of Algorithm 1 can be run on each tensor-shaped parameter group; optimizer APIs in standard deep learning packages promote this convention. This can be viewed as maintaining a tensor sum of preconditioners, each of which is a $p$-wise tensor product.

- The accumulator update step can be implemented concisely and efficiently using re-indexing procedures (typically called `reshape` or `view`) in standard deep learning and numerical linear algebra packages.

Lemma 4.3 in the analysis shows that the per-coordinate learning rates are *underestimates* of those prescribed by AdaGrad. This interpretation is used in our proof, and serves as the basis of the empirical measurements in Section 5.3.

## 4 REGRET ANALYSIS

In this section, we justify the choice of update rules for the compressed preconditioner using the regret-minimizing adaptive regularization framework introduced in the original analysis of Ada-Grad (Duchi et al., 2011). Throughout the analysis, we will adopt the conventions of online convex optimization, as discussed in Section 2.1. This gives rise to an intuitively appealing and interpretable regret bound, which we discuss in this section and revisit in the empirical measurements of Section 5.3.

### 4.1 STATEMENT AND DISCUSSION OF REGRET BOUND

First, we define and prove the main regret bound. As is standard in the regret analysis of diagonal AdaGrad (see Duchi et al. (2011)), this is most naturally stated with a diameter term on $x_t$: $D_\infty := \max_{t,x^*} \|x_t - x^*\|_\infty$. This can be constrained using a projection onto an $\ell_\infty$ ball in the algorithm, but this is seldom seen in practice. It will also be helpful to define $\mathbf{G}_t^i$ to be the diagonal matrix with values

$$\mathbf{G}_t^i[j,j] = \sum_{I:I_i=j} \mathbf{g}_t[I]^2$$

on the diagonal.

**Theorem 4.1.** *Define* $H_T = \otimes_{i=1}^p (\varepsilon \mathbf{I}_{d_i} + \sum_{t=1}^T \mathbf{G}_t^i)^{1/2p}$, *and* $\hat{H}_T = \mathrm{diag}(\varepsilon \mathbf{I} + \sum_{t=1}^T g_t g_t^\top)^{1/2}$. *Then, there exists a choice of constant learning rate schedule* $\eta_1 = \ldots = \eta_t := \eta$ *and* $\varepsilon > 0$ *such that the* $\{x_t\}$ *chosen by Algorithm 1 satisfy the regret bound*

$$\sum_{t=1}^T f_t(x_t) - f_t(x^*) \le D_\infty \sqrt{2\mathbf{Tr}(H_T)\mathbf{Tr}(\hat{H}_T)},$$

*where* $x^* := \min_x \sum_{t=1}^T f_t(x)$ *is the loss-minimizing decision in hindsight.*

As a direct consequence, we recover the AdaGrad regret bound as a special case where $p = 1$. Noting further that $H_T = \hat{H}_T$, we restate this well-known result in our notation, for clarity and comparisons:

**Corollary 4.2.** *In setting of Theorem 4.1, when* $p = 1$, *the* $\{x_t\}$ *chosen by Algorithm 1 satisfy*

$$\sum_{t=1}^T f_t(x_t) - f_t(x^*) \le \sqrt{2} D_\infty \mathbf{Tr}(\hat{H}_T).$$

Thus, AdaGrad with extreme tensoring satisfies a regret bound at most $\sqrt{\mathbf{Tr}(H_T)/\mathbf{Tr}(\hat{H}_T)}$ times than that of AdaGrad. This quantity can be as large as $\Omega(\sqrt{d})$; this is the worst-case price of the dramatic memory savings in the regularizer. However, in the presence of sparsity, this ratio can be much smaller; we include a discussion and empirical measurements in Section 5.3.

### 4.2 PROOFS

The derivation of the regret bound uses the proof framework introduced by Gupta et al. (2017). Our first important lemma states that the regularizer in Algorithm 1 is a spectral (per-coordinate) upper bound for the diagonal AdaGrad preconditioner. In other words, this lemma establishes that the per-coordinate learning rates under extreme tensoring are *underestimates* of those dictated by AdaGrad.

**Lemma 4.3.** *Suppose* $\mathbf{g}_1, \mathbf{g}_2, \cdots, \mathbf{g}_T$ *are tensors of dimension* $d_1 \times \cdots \times d_p$, *and let* $g_t = \mathcal{I}^{-1}(\mathbf{g}_t)$ *for all* $t$, *where* $\mathcal{I}$ *is any tensor index whose image is* $\times_{i=1}^p [d_i]$. *Then for* $t \in [T]$,

$$\mathrm{diag}(\varepsilon \mathbf{I} + \sum_{s=1}^t g_s g_s^\top)^{1/2} \preceq \otimes_{i=1}^p (\varepsilon \mathbf{I}_{d_i} + \sum_{s=1}^t \mathbf{G}_s^i)^{1/2p}.$$

*Proof.* Let $d = d_1 d_2 \cdots d_p$. Let $j \in [d]$ and let $a_1, \ldots, a_p$ be such that $\mathcal{I}(j) = [a_1, a_2, \cdots, a_p]$.

$$\text{diag}(\varepsilon \mathbf{I} + \sum_{s=1}^{t} g_s g_s^\top)[j,j]^p = (\varepsilon + \sum_{s=1}^{t} \mathbf{g}_s[a_1, a_2, \cdots, a_p]^2)^p \le \Pi_{i=1}^p (\varepsilon + \sum_{s=1}^{t} \sum_{I:I_i = a_i} \mathbf{g}_s[I]^2).$$

Taking the $p$th root on both sides,

$$\text{diag}(\varepsilon \mathbf{I} + \sum_{s=1}^{t} g_s g_s^\top)[j,j] \le \Pi_{i=1}^p (\varepsilon + \sum_{s=1}^{t} \sum_{I:I_i = a_i} \mathbf{g}_s[I]^2)^{1/p} = \Pi_{i=1}^p (\varepsilon + \sum_{s=1}^{t} \mathbf{G}_s^i[a_i, a_i])^{1/p}$$

$$= \otimes_{i=1}^p (\varepsilon \mathbf{I}_{d_i} + \sum_{s=1}^{t} \mathbf{G}_s^i)^{1/p}[j,j].$$

Taking square roots of the above inequality yields the lemma. $\square$

### 4.3 REGRET BOUND

The following lemma bounds the regret in terms of the quadratic norms in the regularizer and its dual:

**Lemma 4.4.** *Denote by $H_t$ the $d$-by-$d$ diagonal matrix whose entries are $(\mathcal{I}^{-1}(\delta_t))^{-1}$. Take $\eta_1 = \ldots = \eta_t := \eta$ to be the constant learning rate, and let $x^*$ be the loss minimizing choice in hindsight. Then, the regret of Algorithm 1 is bounded by*

$$\frac{1}{2\eta} \sum_{t=1}^{T} (\|x_t - x^*\|_{H_t}^2 - \|x_{t+1} - x^*\|_{H_t}^2) + \frac{\eta}{2} \sum_{t=1}^{T} (\|g_t\|_{H_t}^*)^2.$$

*Proof.* By the definition of Algorithm 1, for any $x^*$,

$$x_{t+1} - x^* = x_t - x^* - \eta H_t^{-1} g_t, \text{ and } H_t(x_{t+1} - x^*) = H_t(x_t - x^*) - \eta g_t.$$

Taking the inner product of the above vectors,

$$(x_{t+1} - x^*)^\top H_t(x_{t+1} - x^*) = (x_t - x^*)^\top H_t(x_t - x^*) - 2\eta(x_t - x^*)^\top g_t + \eta^2 g_t^\top H_t^{-1} g_t.$$

Rearranging and dividing by $2\eta$, we have

$$g_t^\top (x_t - x^*) = \frac{1}{2\eta}(\|x_t - x^*\|_{H_t}^2 - \|x_{t+1} - x^*\|_{H_t}^2) + \frac{\eta}{2}(\|g_t\|_{H_t}^*)^2.$$

Using convexity of the loss functions and summing over $t$ yields the lemma. $\square$

Next, we present a lemma that is later used to bound the second term in the regret.

**Lemma 4.5.** *(Gupta et al., 2018) Let $g_1, g_2, \cdots, g_T$ be a sequence of vectors, and let $M_t = \text{diag}(\sum_{x=1}^{t} g_s g_s^\top)$. Let $S^+$ denote the set of diagonal positive definite matrices. Given a function $\Phi$ over $S^+$, define $A_t = \arg\min_{A \in S^+} \{M_t \bullet A^{-1} + \Phi(A)\}$ and assume that the minimum is attained for all $t$. Then*

$$\sum_{t=1}^{T} (\|g_t\|_{A_t}^*)^2 \le \sum_{t=1}^{T} (\|g_t\|_{A_T}^*)^2 + \Phi(A_T) - \Phi(A_0).$$

We proceed to prove Theorem 4.1.

*Proof.* (Theorem 4.1) Recall the definition of $H_t = \otimes_{i=1}^p (\varepsilon \mathbf{I}_{d_i} + \sum_{s=1}^{t} \mathbf{G}_s^i)^{1/2p}$, we bound the first term in Lemma 4.4 as follows,

$$\sum_{t=1}^{T} (\|x_t - x^*\|_{H_t}^2 - \|x_{t+1} - x^*\|_{H_t}^2) \le \sum_{t=2}^{T} (x_t - x^*)^T (H_t - H_{t-1})(x_t - x^*) + \|x_1 - x^*\|_{H_1}^2$$

$$\le D_\infty^2 \sum_{t=2}^{T} \mathbf{Tr}(H_t - H_{t-1}) + D_\infty^2 \mathbf{Tr}(H_1) = D_\infty^2 \mathbf{Tr}(H_T)$$

The last inequality is due to the fact that $H_{t-1} \preceq H_t$. Recall $\hat{H}_t = \text{diag}(\varepsilon \mathbf{I} + \sum_{s=1}^{t} g_s g_s^\top)^{1/2}$, and by Lemma 4.3, $\hat{H}_t \preceq H_t$. Now we use Lemma 4.5 with function $\Phi(A) = \mathbf{Tr}(A) + \varepsilon \mathbf{Tr}(A^{-1})$. Let $M_t = \text{diag}(\sum_{s=1}^{t} g_s g_s^\top)$ and let $S^+$ denote the set of diagonal positive definite matrices, we have $\text{argmin}_{A \in S^+}\{M_t \bullet A^{-1} + \Phi(A)\} = \text{argmin}_{A \in S^+} \mathbf{Tr}(\hat{H}_t^2 A^{-1} + A) = \hat{H}_t$. The last equality can be derived by minimizing each diagonal entry of $H$ individually.

By Lemma 4.5,

$$\sum_{t=1}^{T}(\|g_t\|_{\hat{H}_t}^*)^2 \le \sum_{t=1}^{T}(\|g_t\|_{\hat{H}_T}^*)^2 + \Phi(\hat{H}_T) - \Phi(\hat{H}_0) \le \sum_{t=1}^{T}(\|g_t\|_{\hat{H}_T}^*)^2 + \Phi(\hat{H}_T)$$

$$= \text{diag}(\sum_{t=1}^{T} g_t g_t^\top) \bullet \hat{H}_T^{-1} + \mathbf{Tr}(\hat{H}_T) + \varepsilon \mathbf{Tr}(\hat{H}_T^{-1}) = 2\mathbf{Tr}(\hat{H}_T)$$

We can take the diagonal of $\sum_{t=1}^{T} g_t g_t^\top$ in the first equality since $\hat{H}_T^{-1}$ is a diagonal matrix as well.

We proceed to bound the second term in Lemma 4.4. Since $\hat{H}_t \preceq H_t$ and both $H_t$ and $\hat{H}_t$ are full-rank, $\hat{H}_t^{-1} \succeq H_t^{-1}$. Therefore $\sum_{t=1}^{T}(\|g_t\|_{H_t}^*)^2 \le \sum_{t=1}^{T}(\|g_t\|_{\hat{H}_t}^*)^2 \le 2\mathbf{Tr}(\hat{H}_T)$. Summing up the two terms and taking $\eta = D_\infty \frac{\sqrt{\mathbf{Tr}(H_T)}}{\sqrt{2\mathbf{Tr}(\hat{H}_T)}}$, we conclude that the regret of Algorithm 1 is bounded by $\frac{1}{2\eta} D_\infty^2 \mathbf{Tr}(H_T) + \eta \mathbf{Tr}(\hat{H}_T) = D_\infty \sqrt{2\mathbf{Tr}(H_T)\mathbf{Tr}(\hat{H}_T)}$. $\qquad \square$

## 5 EXPERIMENTS

In this section, we provide several empirical studies on extreme tensoring. Our main experiment interpolates between the memorylessness of SGD and the full memory consumption of AdaGrad on a large-scale language model; we additionally isolate the effects of preconditioner expressivity in a synthetic experiment of the same form, and provide a parallel CIFAR-10 experiment in the appendix.

To accompany the main experimental result, we provide empirical measurements of the competitive ratio in the regret bound, as well as a quick comparison in which the memory savings are used to train a larger model.

### 5.1 MEMORY-PERFORMANCE TRADEOFF IN LARGE-SCALE NLP

Our main empirical study focuses on large-scale language modeling with the Transformer architecture (Vaswani et al., 2017) on the Google Billion Words (GBW) dataset (Chelba et al., 2013), and the results are shown in Figure 2. We use the pipeline from the open-source Tensor2Tensor package (Vaswani et al., 2018) for preprocessing the GBW dataset, and use the base Transformer architecture in the same repository as our base architecture. The decoder-only model has 6 identical layers with hidden dimension $d_{model} = 512$, and feedforward dimension $d_{ff} = 2048$. In addition, the weights are shared between the embedding and softmax layers, and the model has a total of $\sim 35M$ parameters.

For our experiments, we use a learning rate schedule of $\eta_t = c \cdot \min(10^{-6} \cdot t, \frac{1}{\sqrt{t}})$, and $c$ is a hyperparameter we tune for each experiment. The learning rate schedule is the same as the one used in (Shazeer and Stern, 2018): a linear warmup stage, followed by inverse square root decay. We train each model for 500K steps on GPUs, with a max sequence length of 256 tokens, and a max number of 4096 tokens in a batch. Global learning rates are selected by hyperparameter search.

Except in our comparison with Adam, we do not use momentum in our extreme tensoring experiments. Momentum incurs an overhead memory cost linear in the model dimension. In addition, we empirically tested the benefit of exponentially decaying the second moment estimator ($\beta_2 < 1$ in the language of Adam and Adafactor). We found that in language modeling experiments, decaying the second moment estimator did not contribute to better performance; on the other hand, the vision experiments in the appendix do use this decay ($\beta_2 = 0.99$).

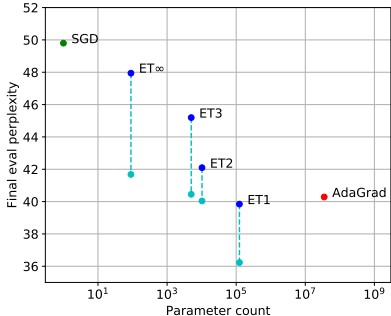

| Optimizer | Parameter count | Final ppl. |
|-----------|-----------------|------------|
| AdaGrad | $3.5 \times 10^7$ | 41.18 |
| ET1 | $1.2 \times 10^5$ | 39.84 |
| ET2 | $1.0 \times 10^4$ | 42.10 |
| ET3 | $5.0 \times 10^3$ | 45.19 |
| ET$\infty$ | 90 | 47.94 |
| SGD | 1 | 49.80 |
| Adam | $7.0 \times 10^7$ | 38.47 |
| Adafactor | $1.2 \times 10^5$ | 38.86 |

Figure 2: Memory-performance tradeoff for GBW language modeling with a Transformer network. Cyan points: performance after doubling model size using optimizer RAM savings.

Table 1: Performance and memory comparison between adaptive optimizers for GBW language modeling with a Transformer network.

Extreme tensoring gives us a class of optimizers which interpolate smoothly between AdaGrad and SGD. Between these endpoints, we choose three levels of extreme tensoring (denoted by ET$\{1, 2, 3\}$), with tensor index dimensions fully specified in the appendix. While extreme tensoring beyond level 2 have limited memory savings in most scenarios, this family of optimizers allow us to have additional interpolating points to investigate the memory tradeoff. The dark blue dots in Figure 2 records the training performance for these interpolating algorithms. Note that the horizontal scale is logarithmic, so extreme tensoring saves memory by orders of magnitude. We also provide training results for Adam (which consumes more memory for first-moment estimation) and Adafactor (similar to ET1 but with a different step size scaling) in Table 1, for completeness; however, these are *not* part of the interpolation study.

To provide an additional interpolating point, we include in the comparison a closely related algorithm, which selects a single learning rate per parameter group (e.g. embedding, bias, convolutional layer); thus, the preconditioner is a tensor sum of scalar multiples of the identity matrix. Each such scalar is chosen to be the inverse square root of the sum of squared $\ell_2$ norms of the parameter group over time; it is easily seen that this achieves the same regret as online gradient descent (Zinkevich, 2003), so we include it as the least granular adaptive optimizer, and call it ET$\infty$. Note that this algorithm is known, and Ward et al. (2018) in particular provide a fine-grained analysis.

Finally, in Appendix 6, we re-run the ET$\{1, 2, 3\}$ experiments with the SM3 update rule (Anil et al., 2019), finding the same monotonic trend. In addition to model performance, we also provide in the appendix wall clock comparisons for the algorithms we have benchmarked.

## 5.2 DOUBLING THE MEMORY ALLOWANCE

As an extension of the above comparison, we argue that the memory consumption freed up by choosing a compressed adaptive optimizer can be usefully reallocated to training a larger model. We double the number of layers of the Transformer network used in the previous section, keeping all embedding sizes the same; this results in a network with $\sim 56$M parameters.

Results are shown in Figure 2 and Table 2. Table 2 reports performance from the larger model, where final perplexities are given with the same running time allowance as the corresponding main experiment (middle column), as well as the same iteration count (500K steps; right column). The light blue dots in Figure 2 plots the final evaluation perplexity of the larger model in the third column of Table 2 for a more intuitive visualization. Holding the memory consumption constant, we find that it pays off to use a larger model rather than an adaptive optimizer that stores all of the second-moment accumulators. Training the larger model with extreme tensoring consumes essentially the same amount of memory as training the original model with Adam or AdaGrad, but the performance improves significantly. Even holding the running time constant, the larger models are competitive with the fully-converged smaller ones.

| Optimizer | Final ppl. *(time)* | Final ppl. *(iters)* |
|-----------|---------------------|----------------------|
| ET1 | 39.25 | 36.23 |
| ET2 | 43.81 | 40.04 |
| ET3 | 44.70 | 40.45 |
| ET∞ | 42.95 | 41.68 |

Table 2: Comparison of memory-efficient optimizers on a double-sized model, so that total memory consumption is lower than that of AdaGrad/Adam on the smaller model. Final perplexities are given with the same running time allowance as the corresponding main experiment (middle column), as well as the same iteration count (500K steps; right column).

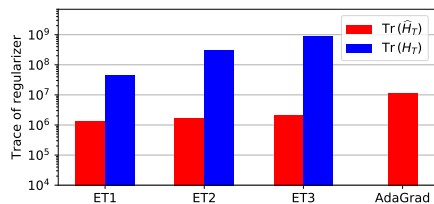

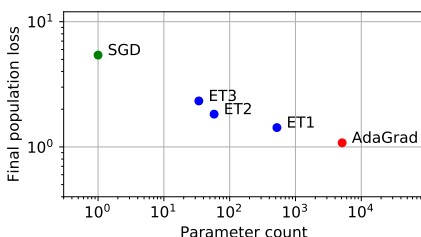

Figure 3: Comparison of quantities in the numerical regret bounds. Since the vertical scale is **logarithmic**, the multiplicative regret bound gap compared to AdaGrad is half the height difference between blue and red bars.

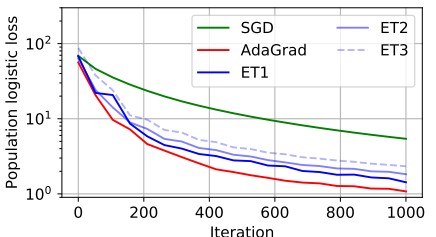

Figure 4: Training curves and final loss comparison for a convex problem with synthetic data. *Left:* Loss curve for each optimizer. *Right:* Final loss vs. optimizer parameter count.

## 5.3 EMPIRICAL MEASUREMENTS OF THE REGRET BOUND

An appealing part of our regret analysis is the interpretability of the trace quantities in Theorem 4.1. Figure 3 shows a comparison of traces $\mathbf{Tr}(H_T)$ and $\mathbf{Tr}(\hat{H}_T)$ of the final regularizers in the language modeling experiment. Then, by Theorem 4.1 and Corollary 4.2, the upper bound for AdaGrad's regret scales with the latter quantity, while that of extreme tensoring scales with the geometric mean of the two traces. Note that the vertical scale of the figure is logarithmic, so the multiplicative regret bound gap compared to AdaGrad is half the height difference between blue and red bars.

Intriguingly, the multiplicative gap, which depends on the loss function and training trajectory via the sequence of gradients, appears to be very small in practice. In the case of one-level extreme tensoring, we have $\sqrt{\mathbf{Tr}(H_T)/\mathbf{Tr}(\hat{H}_T)} \approx 5.7$.

## 5.4 COMPARISON ON SYNTHETIC DATA

In this section, we exhibit a simple experiment with a convex optimization problem, in which there is a clear tradeoff between preconditioner quality and expressivity. We generated Gaussian data $x_i \in \mathbb{R}^{512}$ with a large condition number ($\sim 10^4$), and a Gaussian matrix $W \in \mathbb{R}^{10 \times 512}$. Labels are generated according to the log-linear model $\Pr[y_i = j] \propto \exp((Wx_i)_j)$. Then, the optimization problem in $W$ is to minimize the empirical negative log-probability under the same model. Our findings are robust with respect to the batch size; we use the full gradient on $10^4$ samples in our plots for clarity.

As in the large-scale experiment, we use successively deeper tensor factorizations of the preconditioner, along the feature dimension of the matrix $W$. For depths 1, 2, and 3, we choose tensor indices of dimensions $(10, 512)$, $(10, 16, 32)$, and $(10, 8, 8, 8)$, respectively. Global learning rates are tuned individually. Results are shown in Figure 4.

| Algorithm | Adafactor | Adam | AdaGrad | ET1 | ET2 | ET3 | ET∞ | SGD |
|---|---|---|---|---|---|---|---|---|
| Wall-clock time (hrs) | 20.64 | 18.26 | 17.84 | 18.61 | 20.17 | 24.82 | 19.00 | 17.50 |

Table 3: Wall clock comparisons for optimizing the base Transformer model.

## 5.5 IMPLEMENTATION DETAILS

For completeness, we provide some details on our implementation of the algorithm. Table 3 gives the wall clock comparisons for optimizing the base Transformer model. We point out that the ET algorithms incur some computational overhead due to additional tensor operations. We have not optimized our implementations, and expect that the time efficiency gap could be reduced with architecture-aware optimizations. We trained the model on one V100 GPU, with parallel hyperparameter tuning on the global learning rate multiplier. We used the Tensor2Tensor (Vaswani et al., 2018) framework for training, and Vizier (Golovin et al., 2017) for hyperparameter tuning.

## 6 CONCLUSION

We have introduced *extreme tensoring*, a modification to any second-moment-based adaptive optimizer which drastically shrinks memory overhead. Our experiments characterize a performance-memory tradeoff in the optimizer, and demonstrate the possibility of negligible memory overhead without degrading convergence. Our regret analysis provides a competitive ratio with uncompressed adaptive methods, giving an additional empirical lens on data-dependent preconditioner quality.

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

# A CIFAR-10 EXPERIMENT

In this section, we evaluate the memory-performance trade-off of our proposed algorithm on the CIFAR-10 dataset (Krizhevsky, 2009). Specifically, we compare the test accuracy of a ResNet-18 model (He et al., 2016) trained with SGD, Adam and 3 levels of extreme tensoring.

## A.1 SETUP

With each optimizer, the model is trained for 150 epochs with optimally tuned constant learning rate. To prevent a 2X memory overhead, no momentum is used for any of the optimizers (i.e. $\beta_1 = 0$ in Adam). We use batch size 128, weight decay $5 \times 10^{-4}$ for all the experiments in this section.

## A.2 TENSOR INDICES

We show our tensor decomposition scheme in the table below. Note that bias parameters are not shown in each layer in the table since are treated as vectors and thus not decomposed at all.

| Parameter Shape | Shape (ET1) | Shape (ET2) | Shape (ET3) |
|---|---|---|---|
| $(64, 3, 3, 3)$ | $(64, 3, 9)$ | $(8, 8, 3, 9)$ | $(8, 8, 3, 9)$ |
| $(64, 64, 3, 3)$ | $(64, 64, 9)$ | $(8, 8, 8, 8, 9)$ | $(8, 8, 8, 8, 9)$ |
| $(128, 64, 3, 3)$ | $(128, 64, 9)$ | $(8, 16, 8, 8, 9)$ | $(8, 4, 4, 8, 8, 9)$ |
| $(128, 128, 3, 3)$ | $(128, 128, 9)$ | $(8, 16, 8, 16, 9)$ | $(8, 4, 4, 8, 4, 4, 9)$ |
| $(256, 128, 3, 3)$ | $(256, 128, 9)$ | $(16, 16, 8, 16, 9)$ | $(4, 4, 4, 4, 8, 4, 4, 9)$ |
| $(256, 256, 3, 3)$ | $(256, 256, 9)$ | $(16, 16, 16, 16, 9)$ | $(4, 4, 4, 4, 4, 4, 4, 4, 9)$ |
| $(512, 256, 3, 3)$ | $(512, 256, 9)$ | $(32, 16, 16, 16, 9)$ | $(8, 4, 4, 4, 4, 4, 4, 4, 9)$ |
| $(512, 512, 3, 3)$ | $(512, 512, 9)$ | $(32, 16, 32, 16, 9)$ | $(8, 4, 4, 4, 8, 4, 4, 4, 9)$ |
| $(128, 64, 1, 1)$ | $(128, 64)$ | $(16, 8, 8, 8)$ | $(4, 4, 8, 8, 8)$ |
| $(256, 128, 1, 1)$ | $(256, 128)$ | $(16, 16, 16, 8)$ | $(4, 4, 4, 4, 4, 4, 8)$ |
| $(512, 128, 1, 1)$ | $(512, 128)$ | $(32, 16, 16, 8)$ | $(8, 4, 4, 4, 4, 4, 8)$ |

Table 4: Tensor indices used for different levels of extreme tensoring for the ResNet-18 model on Cifar-10.

## A.3 RESULTS

In this section, we report the best test errors seen in the first 150 epoch along with the parameter count in the optimizer. A similar trend of trade-offs between performance and parameter count is observed.

| Optimizer | Parameter count | Final test error |
|---|---|---|
| Adam($\beta_1 = 0$) | $1.1 \times 10^7$ | 8.40 |
| ET1 | $2.3 \times 10^4$ | 7.22 |
| ET2 | $1.6 \times 10^4$ | 8.49 |
| ET3 | $1.5 \times 10^4$ | 8.52 |
| ET$\infty$ | 62 | 8.57 |
| SGD | 1 | 9.27 |

Table 5: Performance and memory comparison between adaptive optimizers for Cifar-10 classification with a ResNet-18 network.

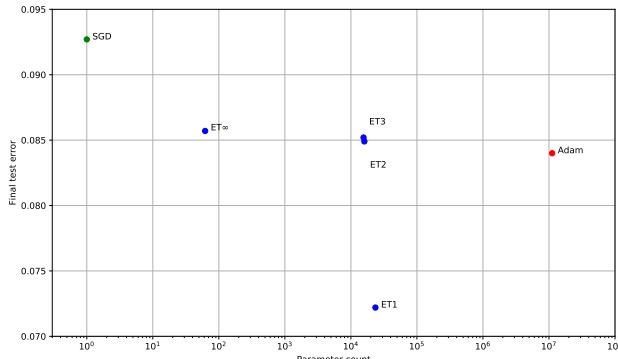

Figure 5: Memory-performance comparison for CIFAR-10 classification with an 18-layer ResNet: final test error vs. optimizer parameter count. Note that the horizontal scale is **logarithmic**.

## B  LANGUAGE MODELING EXPERIMENT DETAILS

In this section, we provide more details on the tensor decomposition in the main empirical result. The following table records parameter shapes for the Transformer model in our main experiments, and the tensor shapes of the reshaped parameters.

| Parameter | Parameter Shape | Shape (ET1) | Shape (ET2) | Shape (ET3) |
|---|---|---|---|---|
| $Q, K, V$, position-wise FF | $(512, 512)$ | $(512, 512)$ | $(16, 32, 16, 32)$ | $(4, 4, 4, 8, 4, 4, 4, 8)$ |
| Embedding Weights | $(2000, 512)$ | $(2000, 512)$ | $(40, 50, 16, 32)$ | $(5, 8, 5, 10, 4, 4, 4, 8)$ |
| Layer Norm | $(512, )$ | $(512, )$ | $(16, 32)$ | $(4, 4, 4, 8)$ |
| FC Weights | $(512, 2048)$ | $(512, 2048)$ | $(16, 32, 32, 64)$ | $(4, 4, 4, 8, 4, 8, 8, 8)$ |
| FC Bias | $(2048, )$ | $(2048, )$ | $(32, 64)$ | $(4, 8, 8, 8)$ |
| FC Weights | $(2048, 512)$ | $(2048, 512)$ | $(32, 64, 16, 32)$ | $(4, 8, 8, 8, 4, 4, 4, 8)$ |
| FC Bias | $(512, )$ | $(512, )$ | $(16, 32)$ | $(4, 4, 4, 8)$ |

### B.1  WALL CLOCK COMPARISONS

In this subsection we provide the wall clock comparisons for optimizing the base Transformer model. We trained the model on one V100 GPU, with parallel hyperparameter tuning on the global learning rate multiplier. We used the Tensor2Tensor (Vaswani et al., 2018) framework for training, and Vizier (Golovin et al., 2017) for hyperparameter tuning.

| Algorithm | Adafactor | Adam | AdaGrad | ET1 | ET2 | ET3 | ET$\infty$ | SGD |
|---|---|---|---|---|---|---|---|---|
| Wall-clock time (hrs) | 20.64 | 18.26 | 17.84 | 18.61 | 20.17 | 24.82 | 19.00 | 17.50 |

## C  SM3 COMPARISON

An alternate way to modulate the memory-performance tradeoff is to use the user-selectable partitions in SM3 (Anil et al., 2019), which provide a alternate scheme for overestimating the AdaGrad preconditioners. These are compatible with the tensor factorizations in our work, so we provide in Table 6 the final test perplexities for the same Transformer experiments on the ET$\{1, 2, 3\}$ decompositions (denoted as S$\{1, 2, 3\}$). We used SM3-II as reported in the pseudocode of Anil et al. (2019). Note that we *have not* attempted to retune any hyperparameters other than the global scalar learning rate multiplier. As such, these experiments should not be viewed as a comparison between these families of optimizers; rather, these experiments corroborate a loss of adaptivity as the preconditioner is constrained to an extremely low dimension.

| Optimizer | Parameter count | Final ppl. |
|-----------|-----------------|------------|
| AdaGrad | $3.5 \times 10^7$ | 41.18 |
| S1 | $1.2 \times 10^5$ | 41.59 |
| S2 | $1.0 \times 10^4$ | 43.25 |
| S3 | $5.0 \times 10^3$ | 49.45 |
| SGD | 1 | 49.80 |

Table 6: Results of Transformer experiments with SM3 preconditioner estimates. The same monotonic behavior is exhibited as when the extreme tensoring granularity is varied.

