# OpenReview forum: "Extreme Tensoring for Low-Memory Preconditioning "
_ICLR.cc/2020/Conference — Accept (Poster)_

### Official Review · AnonReviewer3 · 2019-10-23
**Official Blind Review #3**

**Rating:** 6

**Review:**


============================== Update after rebuttal =======================================================

I did not have any major concerns about the paper in my initial review, only some suggestions for improving the presentation. The authors have addressed most of these issues in their revision. I would like to keep my score as it is. The work seems simple and sound, but somewhat incremental. To have a more meaningful impact, I strongly encourage the authors to make an optimized implementation publicly available as an easy-to-use, plug-and-play type optimizer.

========================================================================================================

This paper proposes a new memory-efficient pre-conditioning scheme for stochastic optimizers. The basic idea is to store a coarse-grained pre-conditioner, expressed as a rank-one tensor product, instead of the full-dimensional pre-conditioner typically used in algorithms like AdaGrad. I am not very familiar with prior work in this literature, but the proposed approach seems simple and sound.

A regret bound is provided for the proposed method, however the analysis here seems to be a straightforward application of the results and techniques from a few prior works. So, I was a bit surprised to see so much space devoted to the proofs. These can be safely moved to the appendix in my opinion. Moreover, the bound does not seem to be very useful in practice. For example, in simulations in Figure 3, the proposed ET1 performs better than AdaGrad, but the bound is not able to capture this at all.

The presentation of the experimental results in section 5 can be improved in my opinion. The authors keep referring to “Figure 5” in this section, but I think this is a typo and these should be “Figure 2” instead. In Figure 2, please indicate on the figure itself what dark blue and light blue colors correspond to (smaller and larger models, respectively).

In Appendix B, wall clock results are presented for different algorithms. These show that the proposed approach is slower than standard algorithms like Adam or AdaGrad. Please explicitly mention this result in the main text (last paragraph of section 5.1) and discuss why this is the case (is this because of the extra reshaping operations required in the updates?).

**Experience Assessment:**

I do not know much about this area.

**Review Assessment: Checking Correctness Of Derivations And Theory:**

I assessed the sensibility of the derivations and theory.

**Review Assessment: Checking Correctness Of Experiments:**

I assessed the sensibility of the experiments.

**Review Assessment: Thoroughness In Paper Reading:**

I read the paper at least twice and used my best judgement in assessing the paper.

---

> ### Author Response · Authors · 2019-11-14
> **Response to R3**
>
> Thanks for the review. We’ve fixed both cosmetic points in the revision.
>
> @Relevance of theory: We believe that the theoretical presentation is still useful to develop the motivation and basic guarantee, a spectral approximation of the uncompressed AdaGrad preconditioner. The numerical regret bounds are to provide some insight on the approximation factor in practice. Nevertheless, we agree that it remains a significant open problem to bridge the gap between convex vs non-convex theory for adaptive optimizers. See response to Reviewer 2.
>
> @Appendix B: Indeed, there is a wall-clock performance gap arising from reshape/reduce operations, which can possibly be improved with a more careful implementation (which may be architecture-specific). We didn’t attempt to optimize the wall-clock time here, beyond providing an implementation that worked on the large-scale NLP setting.

---

> > ### Comment · AnonReviewer3 · 2019-11-14
> > **please mention wall clock results in main text regardless**
> >
> > Thanks for the response. Could you please mention the wall clock results (and the need for extra reshape/reduce operations) in the main text regardless of whether they can be improved with a more careful implementation?

---

> > > ### Author Response · Authors · 2019-11-14
> > > **Done**
> > >
> > > Uploaded another revision. Moved Table 3 & Appendix B.1 to the main paper, and a note on the time overhead.

---

### Official Review · AnonReviewer1 · 2019-10-24
**Official Blind Review #1**

**Rating:** 6

**Review:**

This paper considers the problem of the need for memory-efficient optimizers given the increase in model complexity. They study memory-efficient adaptively preconditioned gradient methods, and see the trade-offs among expressivity and preconditioner quality. They show results on a large-scale NLP model, and show that the memory overhead can be reduced by 3 orders of magnitude, without sacrificing performance.

+ show tradeoff among training convergence and memory in the optimizer.
+ introduce extreme tensoring -- a modification that can be applied to *any* 2nd moment-based adaptive optimizer. It uses a compressed preconditioner.
+ extend diagonal Shampoo to tensor factorization
+ nicely written Related Work.
+ applied their idea to widely used optimizers, such as AdaGrad, Adam, etc.
+ the derived regret bound is only a multiplicative constant from the regret bound of AdaGrad.
+ interesting experiments showing tradeoff among lack of memory (SGD) and full memory (AdaGrad), applied on a real-world machine learning setting of large-scale natural language modeling with Transformers. They show how their extreme tensoring modification can achieve an intermediate ground among lack of memory and full memory.
- I am missing a plot similar to Figure 4 but instead of being on the synthetic data, being on the application of NLP.

Overall, although I do not have a lot of experience in this area, it appears to be that the introduced family of algorithms can be promising as a nice interpolation among SGD-type algorithms and AdaGrad ones. I would personally have preferred to see a wider range of experiments, considering other application scenarios besides NLP, just to see the wide success of the proposed approach. Also, I would urge the authors to spend a bit more text on explaining the algorithm, so that it is easy for practitioners to try it out.

**Experience Assessment:**

I have read many papers in this area.

**Review Assessment: Checking Correctness Of Derivations And Theory:**

I did not assess the derivations or theory.

**Review Assessment: Checking Correctness Of Experiments:**

I assessed the sensibility of the experiments.

**Review Assessment: Thoroughness In Paper Reading:**

I read the paper at least twice and used my best judgement in assessing the paper.

---

> ### Author Response · Authors · 2019-11-14
> **Response to R1**
>
> Thanks for the encouraging review. We’ve fixed the figure reference typo in the revision.
>
> @Large-scale experiments outside NLP: We focus exclusively on the large-scale NLP setting because it’s the one where studying the memory-performance tradeoff is uniquely important. Also, carefully-tuned SGD often achieves state-of-the-art on vision benchmarks; in NLP we can measure more clearly the advantages of adaptivity.
>
> @Practitioner-friendly exposition: To this end, we will release sample code.

---

### Official Review · AnonReviewer2 · 2019-11-02
**Official Blind Review #2**

**Rating:** 8

**Review:**

============= Update after rebuttal

Thanks for the clarifications; I have updated by score and recommend acceptance.

Please do not forget to implement the promise changes in the camera ready version (e.g. about L-BFGS tradeoff work; as well as gap between OCO and non-convex framework).


==============

Motivated from NLP applications where models with billions of parameters are used, this paper proposes a *memory efficient* variant of Adagrad by maintaining a rank-one tensor approximation of the "second-moment" accumulator normally used in Adagrad. From the theoretical side, a simple regret bound is proved which provides an intuitive quantity quantifying the convergence guarantee loss for memory-efficient version (vs. Adagrad), and which is empirically evaluated to be small on a large-scale language modeling task (Section 5.3). From the empirical side, the memory vs. generalization performance tradeoff is evaluated on this language modeling task, showing that similar performance to Adagrad can be obtained with much a smaller (optimization overhead) memory footprint. A convex toy task also indicate a similar result.

I like this paper, I am leaning towards acceptance. The write-up can be improved in a few places (see detailed comment below); but overall, I find the idea refreshing for optimization and the proof is simple and elegant.

The main motivation for the paper is that these large models used in NLP are often making us hit the memory limitation of the hardware just to store the model, and so one then has to tradeoff the size of the model with the memory requirement of the *optimization* algorithm (e.g. one step-size accumulator per dimension for Adagrad). Trying to get the gains of an adaptive preconditioned gradient method but with lower memory footprint thus seems valuable (and Section 5.2 which compares doubling the size of the model + using their memory efficient method vs. original model + Adagrad highlights the gains one can get).

On the other hand, I wish the paper provided a bit more intuition on when the rank-one tensor approximation structure will give good results (e.g. Section 5.3 gives an empirical measure; but what about some simple theoretical examples which give low values, to provide more intuitions?). In particular, the main inequality relating the Adagrad step-size with the "extreme indexing" step-size is arising from the first equation in the proof of Lemma 4.3 on p.5 -- basically one replace one entry of the squared gradient to store with the whole sum over all entries of the squared gradient except a slice. This inequality can be very loose (O(d) in the worst-case), so it is surprising that one could get good approximation ratios much better than the O(sqrt(d)) worst-case (a square-root is taken afterwards), and I wonder what structure yields this.

I also note that Anil et al. (2019) tackles a similar problem with a different approach, and so ideally a more detailed comparison (theoretically and empirically) would be provided (rather than just one sentence as in the current submission), especially since this appeared on arXiv 9 months before the ICLR deadline. But the authors presented it as concurrent work (and it does not appear published anywhere yet) [and a Google search shows the authors had a first version on arXiv only a few weeks after this one, so this seems righs], and so I decided to not hold it against this submission. Disclaimer: I made my general opinion about the paper before doing these Google searches and finding the paper on arXiv.

I am not very familiar with the modern transformer architecture experiments, so cannot evaluate their quality, but they seem sensible from an outsider's perspective.

== Detailed comments ==

- p.1 "the first empirical study of the tradeoff between training convergence and memory in the optimizer." This should be properly qualified by "in the Adagrad setup" or something like that; I am pretty sure it is false *for general optimizers*. For example, I guess there are several papers which empirically studied the memory - performance tradeoff for L-BFGS optimizers...

- Important: I think the paper should be more transparent about the *big gap* between the online *convex* optimization framework and stochastic optimization for a *non-convex* loss. There is only one sentence at the end of Section 2.1 mentioning vaguely that Agarwal et al. (2018) provide a reduction from stochastic non-convex setting to the online convex setting, but this is buried in the appendix of the said paper, with several caveats, and for a modified algorithm (i.e. as far as I understand, no convergence guarantee would be given for Algorithm 1 to find a stationary point on a non-convex loss; but one could apply Algorithm 1 on a series of convex problems to obtain overall guarantees on the non-convex loss [but this is a different algorithm!]). I suggest that either the argument of Agarwal et al. (2018) is summarized in a few sentences at the end of Section 2.1 to clarify the real link; or give more caveats [also just before 4.1] (e.g., my current guess is that we use the convex analysis setup to gain insights on the behavior in a controlled setup; and then just hope that some of it applies in the non-convex setup, even though no guarantees is provided whatsoever).

- Some undefined notation: line 8 of Algorithm 1, it seems they use the dot for the Hadamard product between two vectors. The big dot in Lemma 4.5 is most likely the dot product between matrices (but please mention explicitly for clarity).

- Typos in Section 5.1 & 5.2: several references seem wrong. E.g. Figure 5 is probably Figure 2; Table 4 is Table 1, etc... Please correct!

- Clarity: I suggest to put much more description in the captions of Figure 2, Table 2, etc. (for example it was much clearer in their arXiv version). I was very confused by the extra light blue dot in Figure 2 which is only explained in 5.2; I suggest that it is already mentioned (with forward pointer) in the caption. Use more of p.9 for the clarity sake...

- Appendix B.1: please provide hardware information when you give wall-clock comparison.

**Experience Assessment:**

I have published in this field for several years.

**Review Assessment: Checking Correctness Of Derivations And Theory:**

I carefully checked the derivations and theory.

**Review Assessment: Checking Correctness Of Experiments:**

I assessed the sensibility of the experiments.

**Review Assessment: Thoroughness In Paper Reading:**

I read the paper thoroughly.

---

> ### Author Response · Authors · 2019-11-14
> **Response to R2**
>
> @Camera-ready update: Done. We replicated the results for SM3, and incorporated the very helpful suggestions for discussing how this fits in with other literature.
>
> ===
>
> Thanks for the particularly thoughtful review. We have incorporated the minor points into the current revision. Responses to the major points below:
>
> @Tensor approximability: Our original intuition for low-rank tensor preconditioning came from the ubiquitous use of SVD for compression. Unlike SVD, it is difficult to usefully characterize the “beyond worst-case” gap between the presented tensor approximation and the true preconditioner. Indeed, if there are sparsity structures present inside a particular tensor slice, the part of the O(d) gap attributed to that slice can be reduced to O(sparsity). One could potentially theorize such sparse models of the data and architecture. Instead, we take the route of measuring this “gap” empirically, and see that it is indeed far from worst-case. By now there is strong evidence that such approximations are useful for preconditioners, and it is an interesting future direction to characterize precisely what underlying structures enable this.
>
> @SM3: Indeed, the algorithm proposed by [Anil et al. ‘19] provides another flexible way to do low-memory preconditioning. Since our main emphasis is on an empirical study of the memory vs performance tradeoff and SM3 also enables this, we would be happy to run our tradeoff experiment with SM3 as well. An important point to note is that ET_\infty, a special case of our algorithm but not SM3, is studied in its own right as “adaptive OGD” (see, e.g. [3]).
>
> @L-BFGS memory comparison: We were not aware of these works on the L-BFGS memory-performance tradeoff, and will add a note in the paper. Since quasi-Newton methods work in a totally different regime than adaptive preconditioning (they tend to behave poorly in typical deep learning settings due to stochasticity), the methods/experiments/conclusions appear to be orthogonal.
>
> @OCO vs non-convex: We are very sympathetic to this point. Connecting insights and theorems from {online, stochastic} convex optimization to their effectiveness (or lack thereof) in the modern deep learning setup remains a major research program in this space (see, e.g. [1,2,3,4]), to which there has not yet been a totally satisfying account. We will add some more comments on some theoretical approaches to bridge the gap. However, we are purposefully not trying to get into the larger debate in this work, instead focusing on algorithms and empirics.
>
> [1] Escaping Saddle Points with Adaptive Gradient Methods. Staib et al., ICML ‘19.
> [2] Optimal Adaptive and Accelerated Stochastic Gradient Descent. Deng et al., arXiv ‘18.
> [3] AdaGrad stepsizes: Sharp convergence over non-convex landscapes, from any initialization. Ward et al., ICML ‘19.
> [4] On the Convergence of Adaptive Gradient Methods for Nonconvex Optimization. Zhou et al., arXiv ‘18.

---

### Decision · Program_Chairs · 2019-12-19

**Decision:**

Accept (Poster)

**Comment:**

Post author rebuttal the score of this paper increased.
Discussions with reviewers were substantive and the AC recommends acceptance.